# Compositional Generative Modeling from Decentralized Data

**Mashrur M. Morshed** [1]   **Vishnu Naresh Boddeti** [1]

## Abstract

Learning the compositional nature of the physical world requires joint observation of interacting factors. However, because practical data is often decentralized, these factors are fragmented across isolated silos. Existing decentralized generative approaches focus only on modeling the *union* of siloed data, overlooking *novel combinations* implied by the collective whole. To bridge this gap, we introduce Decentralized Compositional Flow Matching (DCFM), a framework that enforces structural constraints across the global set of generative factors, without exchanging any raw data. DCFM enables novel combinations to emerge through peer interactions, even when no single data source can independently support the composition. Empirically, DCFM substantially outperforms federated learning and mixture-of-experts baselines across conditional image generation, robotic spatial planning, and medical attribute co-occurrence modeling.

## 1. Introduction

Consider learning a generative model for outdoor robot navigation from data collected by multiple robots deployed under different environmental conditions. Due to communication constraints or operational isolation, each robot stores experience locally and cannot share raw trajectories. One robot observes navigation patterns exclusively in rain, another only in windy conditions, and a third only across varied terrain types (flat roads, off-road) (Figure 1). Although each robot's data captures valid aspects of the navigation task, many operationally critical scenarios are never observed by any single robot during training: navigating flat terrain during combined wind and rain, handling off-road conditions in rain, managing wind on uneven ground, or simultaneously contending with all three factors. Yet these

[1]Department of Computer Science and Engineering, Michigan State University, East Lansing, MI, USA. Correspondence to: Mashrur M. Morshed <morshedm@msu.edu>.

*Proceedings of the $43^{rd}$ International Conference on Machine Learning*, Seoul, South Korea. PMLR 306, 2026. Copyright 2026 by the author(s).

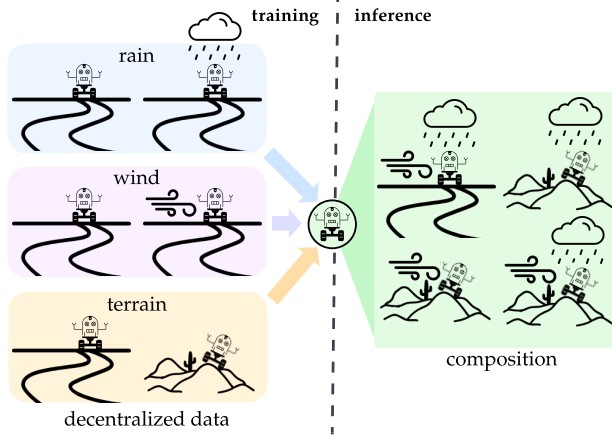

*Figure 1.* (Left) Robots learn to navigate in isolated, specialized silos that model rain, wind, and terrain variation respectively. (Right) The goal is to combine knowledge from the decentralized robots to handle novel, unobserved combinations of environmental factors.

multi-factor compositions are precisely the conditions that deployed robots must handle reliably in the real-world.

Multiple solutions can be employed to learn from decentralized silos. For example, federated learning (McMahan et al., 2017) can be employed to learn a global generative model without sharing data (Augenstein et al., 2020; Tun et al., 2023). Alternatively, mixture-of-experts (McAllister et al., 2025; Hahn & Lee, 2025) or product-of-experts (Zhang et al., 2026) can be used to compose independent locally trained generative models at inference time, leveraging the strengths of each expert. When the data distributions are heterogeneous, the latter strategy can, in theory, be effective and yield strong empirical performance.

However, as we demonstrate in this paper, when the factors required for composition are distributed across isolated data sources, the above solutions break down. Federated learning implicitly assumes that averaging parameters or gradients produces a coherent global model, an assumption violated when local datasets contain disjoint or highly skewed combinations of generative factors. Expert-based approaches, on the other hand, lack structural guarantees (e.g. a shared latent space) for how independently trained models should interact. As a result, novel compositions, such as trajectory patterns that require combining knowledge held by different robots, remain inaccessible, even though all required components are present across the population.

This raises a central research question. *How can generative models recover compositional structure when no single data source contains sufficient information to support composition on its own, and raw data cannot be shared?*

We present Decentralized Compositional Flow Matching (DCFM) to learn generative models that are explicitly designed for compositional generalization from decentralized data without sharing raw data. The key idea is to satisfy conditional independence (CI) globally across data silos when the total coverage of attribute combinations is limited. We introduce two variations of DCFM. The first, DCFM-A, optimizes the global CI objective directly on real local datasets. It preserves a set of decentralized "experts" that maintain high empirical fidelity, though it remains computationally expensive during sample generation. The second, DCFM-B, is an efficient alternative that distills the collective knowledge of the local experts into a monolithic student model through synthetic data replay. By training on composite paths generated by the local flows, the student learns the global CI relations, enabling efficient sampling of novel attribute combinations never seen by any individual expert.

**Summary of contributions.**

1. We demonstrate that standard flow models trained independently suffer from (1) spurious correlation of independent attributes in the local models, and (2) incompatible velocity fields for composition, collectively limiting compositional generalization across isolated data silos. (§ 4)

2. We present the first framework to enable compositional generalization from decentralized data, introducing a training objective that enforces cross-expert compatibility via a shared velocity field and global conditional independence across the data silos. (§ 5)

3. We adopt this objective to learn both mixture-of-experts and a monolithic model. We demonstrate the effectiveness and versatility of DCFM for compositional generalization across conditional image generation and robotic path planning tasks. (§ 6)

## 2. Related Work

**Federated Learning for Diffusion Models.** Federated learning (FL) offers a promising framework for training generative models in privacy-sensitive domains by avoiding centralized data aggregation. Recent works have adapted diffusion training to this setting through federated averaging (Tun et al., 2023) or decentralized sampling protocols (Hahn & Lee, 2025). Despite this progress, practical adoption is hindered by the high computational and communication overheads inherent to the iterative training of large-scale diffusion models (Vora et al., 2024; de Goede

et al., 2024). Furthermore, existing FL methods exhibit limited robustness to non-IID data distributions, often necessitating privacy-weakening workarounds such as partial data sharing (Jothiraj & Mashhadi, 2024) or frequent local retraining (Peng et al., 2025). Crucially, these approaches typically aim to approximate a single global distribution; they are not designed to exploit the specialized expertise within individual data silos, which limits both generation quality and compositional flexibility.

**Mixture of Experts and Compositional Generalization.** Prior work on compositional generalization relies on two critical, often unmet assumptions. In monolithic frameworks that combine conditional scores (Du et al., 2020; Liu et al., 2022; Luo et al., 2025), the validity of composition hinges on the factors being statistically independent. Gaudi et al. (2025) demonstrated that standard training violates this condition and proposed CoInD to explicitly enforce independence. Conversely, mixture-of-experts approaches (McAllister et al., 2025; Zhang et al., 2026; Hahn & Lee, 2025) assume that independently trained models occupy compatible score spaces. While CoInD resolves the independence issue, it operates on centralized data and does not address the compatibility gap inherent to decentralized learning. When experts are trained on isolated data silos, their underlying score fields are not naturally aligned; this incompatibility renders standard composition techniques invalid, resulting in significantly degraded generative performance (e.g., FID) compared to centralized models trained on pooled data. Thus, achieving compositional generalization across decentralized users requires a framework that ensures both factor independence and cross-expert compatibility.

## 3. Problem Statement and Preliminaries

### 3.1. Problem Formulation

Let $\mathbf{x} \in \mathcal{X}$ be a sample associated with an attribute vector $\mathbf{y} = (y_1, y_2, \ldots, y_k) \in \mathcal{Y}$, where $\mathcal{Y}$ is the Cartesian product of discrete spaces, $\mathcal{Y} = \mathcal{Y}_1 \times \mathcal{Y}_2 \times \cdots \times \mathcal{Y}_k$. We assume the attributes are conditionally independent given $\mathbf{x}$, meaning their joint conditional distribution factorizes as:

$$p(\mathbf{y} \mid \mathbf{x}) = \prod_{i=1}^{k} p(y_i \mid \mathbf{x}) \tag{1}$$

**Decentralization.** Observations of $\mathbf{x}$ are partitioned across $n$ clients, yielding localized datasets $\mathcal{D} = \{D_1, \ldots, D_n\}$ subject to the following constraint:

*Constraint 1. Any $\mathbf{x} \in D_a$ is strictly accessible only to client $c_a$, ensuring $D_a \cap D_b = \emptyset$ for all $a \neq b$.*

Crucially, Constraint 1 restricts only the exchange of raw samples $\mathbf{x}$. The total attribute space $\mathcal{Y}$ is globally known; every client $c_a$ is aware of all possible attribute classes.

**Coverage.** Because compositional capability is reliant on the observation of compositional factors, we formalize the notion of *coverage* of a dataset. The **total coverage** $\mathcal{C} \in [0, 1]$ denotes the fraction of *observed combinations* of the product space $\mathcal{Y}$, while the **marginal coverage** $\mathcal{M}_i \in [0, 1]$ is the fraction of *observed outcomes* for a specific $\mathcal{Y}_i$.

Following Wiedemer et al. (2023), compositional generalization strictly requires full global marginal coverage ($\mathcal{M}_i = 1$ for all $i$), ensuring every attribute outcome exists somewhere in the system. However, merely satisfying $\mathcal{M}_i = 1$ is insufficient for the model to isolate the effect of individual attributes. We thus define a strict lower bound on total coverage necessary for generalization.

**Definition 3.1** (Minimum Compositional Coverage). To enable attribute disentanglement and support compositional generalization, the total coverage $\mathcal{C}$ must satisfy:

$$\mathcal{C} \geq \mathcal{C}_{\min} = \frac{1 + \sum_{i=1}^{k}(|\mathcal{Y}_i| - 1)}{|\mathcal{Y}|} \quad (2)$$

The numerator in Eq. (2) represents the minimum number of unique joint observations required to satisfy all main-effect degrees of freedom in a compositional model. Moving forward, we assume $\mathcal{D}$ operates in a feasible regime satisfying both $\mathcal{M}_i = 1$ globally and $\mathcal{C} \geq \mathcal{C}_{\min}$.

**Problem.** Let $G_\theta : \mathcal{Z} \times \mathcal{Y} \rightarrow \mathcal{X}$ be a generative model that maps a latent variable $\mathbf{z} \in \mathcal{Z}$ and an attribute vector $\mathbf{y} \in \mathcal{Y}$ to the data space. Our goal is to learn the parameters $\theta$ over the decentralized data $\mathcal{D}$ such that $G_\theta$ can sample from $p(\mathbf{x} \mid \mathbf{y})$, even for attribute combinations $\mathbf{y} \in \mathcal{Y}$ that were unobserved during training (i.e., $\mathbf{y} \notin \mathcal{D}$). We formalize the inference objective as:

> *Problem* 1.
>
> $$\max_{\theta} \mathbb{E}_{\mathbf{z} \in \mathcal{Z}, \mathbf{y} \in \mathcal{Y}} \left[ \prod_{i=1}^{k} \mathbb{1} \left\{ G_\theta(\mathbf{z}, \mathbf{y}) \in \mathrm{supp}(p(\mathbf{x} \mid y_i)) \right\} \right]$$
>
> s.t. Constraint 1,

where the indicator function $\mathbb{1}\{\cdot\}$ enforces that the generated sample $G_\theta(\mathbf{z}, \mathbf{y})$ must reside within the intersection of the supports of the marginal conditionals $p(\mathbf{x}|y_i)$ for all $y_i \in \mathbf{y}$. Thus, $G_\theta$ must achieve **compositional generalization** by correctly composing independent attribute representations learned from decentralized, sparse data.

### 3.2. Flow Matching and Composition

We interpret $G_\theta$ as a flow matching model (Lipman et al., 2023; Liu et al., 2023), a widely used family of state-of-the-art generative models that are strongly connected to diffusion models (Sohl-Dickstein et al., 2015; Ho et al., 2020) and score-based generative models (Song & Ermon, 2019;

Song et al., 2021). Flow matching learns a time-dependent velocity field $\mathbf{v}_\theta(\cdot, t)$ that generates a probability path to transform a source distribution $p_0$ to a target distribution $p_1$. We can define $G_\theta(\cdot)$ as the integration

$$G_\theta(\mathbf{z}, \mathbf{y}) := \mathbf{x}_0 + \int_0^1 \mathbf{v}_\theta(\mathbf{x}_t, t, \mathbf{y}) dt \quad (3)$$

where $\mathbf{x}_0 \sim \mathcal{N}(\mathbf{0}, \mathbf{I})$ is equivalent to the latent noise variable $\mathbf{z}$. For brevity, we omit $t$ and use $\mathbf{v}_t(\mathbf{x})$ or $\mathbf{v}_\theta(\mathbf{x}_t)$ to imply $\mathbf{v}_\theta(\mathbf{x}_t, t)$. We also use the notation $\mathbf{v}_\theta^{(a)}$ to indicate a model local to some client $c_a$ with distinct local parameters $\theta_a$, and sole access to the dataset $D_a \in \mathcal{D}$.

**Compositional sampling.** Consider a novel condition vector $\mathbf{y} = (y_1, y_2, \ldots, y_k) \notin \mathcal{D}$ but with known marginal attributes $y_i \in \mathcal{D} \, \forall i$. Previous works on diffusion (Liu et al., 2022; Ajay et al., 2023; Gaudi et al., 2025) sample from the product of marginals as

$$\hat{\epsilon}_\theta(\mathbf{x}_t, \mathbf{y}) = \epsilon_\theta(\mathbf{x}_t) + \sum_{i=1}^{k} w_i \left( \epsilon_\theta(\mathbf{x}_t, y_i) - \epsilon_\theta(\mathbf{x}_t) \right) \quad (4)$$

where $w_i$ is analogous to the classifier-free guidance (Ho & Salimans, 2022) strength per attribute. However, the compositional sampling procedure in Eq. (4) has not been explored in the context of flow matching velocities. Therefore, we first derive a *compositional velocity field*, and show that Eq. (4) has the following straightforward extension:

$$\hat{\mathbf{v}}_\theta(\mathbf{x}_t, \mathbf{y}) = \mathbf{v}_\theta(\mathbf{x}_t) + \sum_{i=1}^{k} w_i \left( \mathbf{v}_\theta(\mathbf{x}_t, y_i) - \mathbf{v}_\theta(\mathbf{x}_t) \right) \quad (5)$$

*Sketch of proof of* (5). We can start from the joint conditional score, $\nabla \log p_t(\mathbf{x} \mid \mathbf{y})$, which factorizes as $\nabla \log p_t(\mathbf{x}) + \sum_{i=1}^{k}(\nabla \log p_t(\mathbf{x} \mid y_i) - \nabla \log p_t(\mathbf{x}))$ by assuming Eq. (1) holds. Eq. (4) is derived from this factorization, through the relationship between the diffusion $\epsilon$ and the score $\nabla \log p$. In a similar manner, we can derive the compositional velocity from the compositional score by using the following identity, given by Lemma 1 of Zheng et al. (2023):

$$\mathbf{v}_t(\mathbf{x}_t \mid \mathbf{y}) = a_t \mathbf{x}_t + b_t \nabla \log p_t(\mathbf{x}_t \mid \mathbf{y}) \quad (6)$$

where $a_t, b_t$ are related to the schedule of the probability path $p_t$ between source $p_0 \sim \mathcal{N}(\mathbf{0}, \mathbf{I})$ and target $p_1 \sim p_{\text{data}}$. Full derivation shown in § A.2.

**Enforcing Conditional Independence.** Both Liu et al. (2022) and Ajay et al. (2023) state that Eq. (4) assumes CI (Eq. (1)), while Gaudi et al. (2025) explicitly enforce CI through a penalty. To enforce compositional structure when $\mathcal{C}$ is sparse, we similarly present a CI penalty over $\mathbf{v}$,

$$\mathcal{L}_{\text{CI}}(\theta) = \mathbb{E}_{t, \mathbf{x}_t, \mathbf{y} \sim D_a} \|\mathbf{v}_\theta(\mathbf{x}_t, \mathbf{y}) - \hat{\mathbf{v}}_\theta(\mathbf{x}_t, \mathbf{y})\|^2 \quad (7)$$

where $\hat{\mathbf{v}}_\theta$ follows Eq. (5), with $w_i = 1, \forall i$. By minimizing the discrepancy between the joint velocity and the marginal sum, we encourage flow models to learn a disentangled field that generalizes to unobserved compositions.

## 4. Are Decentralized Flows Compositional?

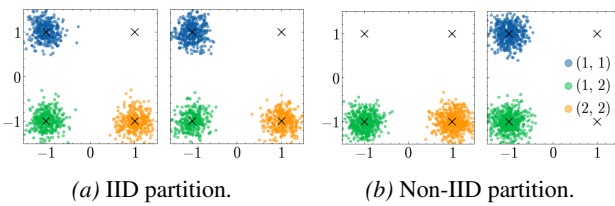

*(a)* IID partition.       *(b)* Non-IID partition.

*Figure 2.* A set of two decentralized private datasets, $\{D_1, D_2\}$, where each point is associated with an attribute vector $\mathbf{y} \in \{1, 2\}^2$.

We elucidate Prob. 1 with a simple example consisting of a mixture of Gaussian distributions. Suppose we have a set of two decentralized private datasets, $\mathcal{D} = \{D_1, D_2\}$. Each data point $\mathbf{x}$ in $\mathcal{D}$ is associated with a horizontal attribute $\mathcal{Y}_1 \in \{\text{left}, \text{right}\}$ and a vertical attribute $\mathcal{Y}_2 \in \{\text{top}, \text{bottom}\}$. By assigning a numeric label to each class, the total attribute space becomes $\mathcal{Y} \in \{1, 2\}^2$. Importantly, the particular combination $\mathbf{y} = (2, 1)$ does not occur in $\mathcal{D}$.

Though $\mathcal{D}$ can be arbitrarily partitioned, we show two informative configurations: (i) an IID partition, where both nodes possess all three available modes, and (ii) a non-IID partition, where both $D_1$ and $D_2$ observe two modes each, as shown in Fig. 2. We adopt three decentralized baselines to learn a flow matching model on $\mathcal{D}$ (results in Fig. 3).

**(i) Federated Flow.** We train a standard flow matching model $\mathbf{v}_\theta$, with iterations of parameter averaging, following Tun et al. (2023). We find that, for non-IID partition, the federated flow shows poor convergence. Further, even under IID partition, the model cannot recover the missing mode.

**(ii) Decentralized Diffusion Models (DDM)** (McAllister et al., 2025) trains an independent flow matching model on each node $a$, $v_\theta^{(a)}$, with no communication with any other node $b \neq a$ during training. Although McAllister et al. (2025) perform a data clustering step before training, we avoid this step as it involves breaking the locality assumption in Constraint 1. After training, the models are shared to some central server (or transmitted to each other) for inference. We construct a lightweight router for the set of models using marginal label statistics. From Fig. 3, we observe that that DDM shows good recovery of the collective *observed* decentralized data, in either partitioning. However, it struggles to generate the missing top-right mode.

**(iii) Diffusion Federated Dataset (DFD)** (Hahn & Lee, 2025) involves a similar process of training independent local experts as DDM. However, instead of simply using a

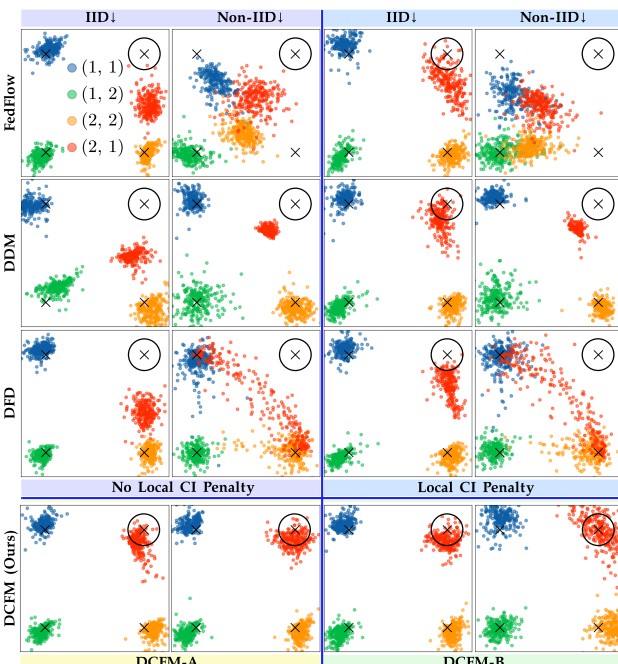

*Figure 3.* **Left two columns:** Prior methods fail to recover the unobserved $\mathbf{y} = (2, 1)$ mode (circled). **Right two columns:** Local conditional independence does not help in non-IID case. **Bottom:** DCFM recovers the missing mode in both IID and non-IID cases.

router, DFD additionally estimates the energy (unnormalized densities) to approximate an ideal weight factor for each expert. Similar to DDM, DFD shows good coverage of the known regions in $\mathcal{D}$. However, under non-IID settings, the energy-based routing system fails for novel attribute composition, showing a tendency to shift samples towards *observed* regions of the data density.

**Why these models fail.** These decentralized models fail to recover the missing mode not only due to the challenges posed by decentralization, but also because the conditional independence (Eq. (1)) assumption does not hold. If the compositional generalization failure stems from the lack of conditional independence in the models, one may naturally pose the question: *Can we simply enforce a conditional independence penalty* (7) *during local training of the flow velocities, to help recover the missing mode?*

**The answer: not necessarily.** Even if every local $p^{(a)}(\mathbf{y} \mid \mathbf{x})$ are conditionally independent, we do not have guarantees that the global mixture $p^*(\mathbf{y} \mid \mathbf{x}) = \sum_{a=1}^n w_a p^{(a)}(\mathbf{y} \mid \mathbf{x})$ satisfies Eq. (1). We observe in Fig. 3 that adding a CI penalty *can be helpful* under IID conditions, where $p^{(a)} \sim p^{(b)}$ for any pair of clients $c_a$, $c_b$ (all clients share similar data distributions). However, when the data are non-IID, simply enforcing local conditional independence cannot solve the compositional problem (Fig. 3, top-right quadrant).

Our approach, DCFM, is primarily concerned with how to learn *global* conditional independence, over the collective

attribute space of all the clients. As shown in Fig. 3, DCFM enables mode recovery under both IID and non-IID settings.

# 5. Decentralized Compositional Flow Models

We now present two variants of DCFM corresponding to an idealized mixture-of-experts and a more practically efficient monolithic model. Both variants share a first stage of learning local models trained on local data. An overview of DCFM is shown in Fig. 4.

## 5.1. Stage I: Local Matching

Similar to mixture-of-expert methods like DDM and DFD, we first optimize local models $\mathbf{v}_\theta^{(a)}$ in order to sufficiently learn the distribution of locally available data. This phase is similar to standard FM training with two minor modifications: (1) marginal label learning, and (2) local CI penalty.

**Marginal labels.** We observe from (4) and (5) that inference utilizes marginal labels, $y_i$. However, during standard training, flow matching models usually have access to full joint labels $\mathbf{y}$ and unconditional labels $\{\emptyset\}^k$. Thus, with a small probability $p_{\mathrm{marg}}$, we find it helpful to introduce marginal labels $(\emptyset, \ldots, y_i, \ldots, \emptyset)$ to the model.

Let $\mathbf{m} \in \{0, 1\}^k$ be a random binary masking vector, drawn from some distribution $p(\mathbf{m})$. We then construct a masked attribute vector $\mathbf{y_m} = \mathbf{y} \odot \mathbf{m}$, where an attribute $y_i$ is replaced by a "null" token $\emptyset$ if $m_i = 0$. We define $p(\mathbf{m})$ as:

$$p(\mathbf{m}) = \begin{cases} \pi_{\mathrm{full}} & \text{if } \mathbf{m} = \mathbf{1} \\ \pi_{\mathrm{marg}} \cdot \frac{1}{k} & \text{if } \mathbf{m} = \mathbf{e}_i, \text{ for } i \in \{1, \ldots, k\} \\ \pi_{\mathrm{uncond}} & \text{if } \mathbf{m} = \mathbf{0} \end{cases} \tag{8}$$

$\pi_{\mathrm{full}}, \pi_{\mathrm{marg}}, \pi_{\mathrm{uncond}} \in [0, 1]$ are mixture weights that sum to 1. The local training objective becomes:

$$\mathcal{L}_{\mathrm{FM}}^{(a)}(\theta) = \underset{\substack{t, \mathbf{y}, \mathbf{m} \\ \mathbf{x}_t \sim p_t}}{\mathbb{E}} \left[ \left\| \mathbf{v}_\theta^{(a)}(\mathbf{x}_t, t, \mathbf{y} \odot \mathbf{m}) - \mathbf{u}_t(\mathbf{x}_t | \mathbf{x}_1) \right\|^2 \right] \tag{9}$$

where $t \sim \mathcal{U}[0, 1]$ is the timestep, and $\mathbf{u}_t(\mathbf{x}_t \mid \mathbf{x}_1)$ is the ground-truth conditional velocity field, which for a linear path is simply $\mathbf{x}_1 - \mathbf{x}_0$ (where $\mathbf{x}_0 \sim \mathcal{N}(\mathbf{0}, \mathbf{I})$ and $\mathbf{x}_1 \in D_a$).

**Local CI.** We adopt the local CI penalty (7) when the full labels are available. If $\hat{\mathbf{v}}$ denotes the composition of marginal velocities (5), we define the local penalty as:

$$\mathcal{L}_{\mathrm{CI}}^{(a)}(\theta) = \underset{t, \mathbf{y}, \mathbf{x}_t \sim p_t}{\mathbb{E}} \left[ \mathbf{v}_\theta^{(a)}(\mathbf{x}_t, \mathbf{y}) - \hat{\mathbf{v}}_\theta^{(a)}(\mathbf{x}_t, \mathbf{y}) \right] \tag{10}$$

The total local loss becomes

$$\mathcal{L}_{\mathrm{total}}^{(a)}(\theta) = \mathcal{L}_{\mathrm{FM}}^{(a)}(\theta) + \lambda \cdot \mathcal{L}_{\mathrm{CI}}^{(a)}(\theta) \tag{11}$$

where $\lambda$ denotes the strength of the CI penalty.

**Model aggregation.** After Stage I, each client shares their model with either a trusted server, or with other clients.

## 5.2. DCFM-A: Learning Local Experts With Cross-Peer Conditional Independence.

We consider that the set of clients perform an `allgather` operation over $\theta$, and that each client now has access to a set of local experts, $\{\mathbf{v}_\theta^{(a)}\}_{a=1}^n$. Next, let $\mathbf{r} = (r_0, r_1, r_2, \ldots, r_k)$ be a routing vector, where any $r_i \in \{1, 2, \ldots, n\}$. For $i = 1 \ldots k$, $r_i$ assigns a model for each of the $k$ marginal attributes $\{y_i\}_{i=1}^k$, while $r_0$ is a model choice for the unconditional attribute $\emptyset$. The routing vector $\mathbf{r}$ allows us to redefine the conditional independence penalty (10) over an arbitrary combination of models as,

$$\mathbf{z}_\theta^{(r_0)}(\mathbf{x}_t) = \bar{\mathbf{v}}_\theta^{(r_0)}(\mathbf{x}_t) + \sum_{i=1}^k \left( \bar{\mathbf{v}}_\theta^{(r_i)}(\mathbf{x}_t, y_i) - \bar{\mathbf{v}}_\theta^{(r_0)}(\mathbf{x}_t) \right)$$

$$\mathcal{L}_{\mathrm{peerCI}}^{(a)}(\theta) = \underset{\mathbf{r}, t, \mathbf{y}, \mathbf{x}_t \sim p_t}{\mathbb{E}} \left[ \mathbf{v}_\theta^{(a)}(\mathbf{x}_t, \mathbf{y}) - \mathbf{z}_\theta^{(r_0)}(\mathbf{x}_t) \right] \tag{12}$$

where $\bar{\mathbf{v}}_\theta^{(\cdot)}$ is a frozen model when $r \neq a$ and is defined as

$$\bar{\mathbf{v}}_\theta^r = \delta_{r,a} \mathbf{v}_\theta^{(a)} + (1 - \delta_{r,a}) \mathrm{StopGrad}(\mathbf{v}_\theta^{(r)}). \tag{13}$$

where $\delta_{i,j}$ is the Kronecker delta. That is, we only update the parameters of the local model $\mathbf{v}_\theta^{(a)}$ while keeping the peer experts frozen. Further, any assignment $\mathbf{r}$ ensures that there is at least a single $r_i = a$. The total loss is now,

$$\mathcal{L}_{\mathrm{DCFM-A}}^{(a)}(\theta) = \mathcal{L}_{\mathrm{FM}}^{(a)}(\theta) + \lambda \cdot \mathcal{L}_{\mathrm{peerCI}}^{(a)}(\theta) \tag{14}$$

Thus, (14) trains local experts $\mathbf{v}_\theta^{'(a)}$ that learn conditional independence both locally and across peers. As seen in Fig. 3 (bottom), sampling from a mixture of DCFM-A experts shows compositional generalization even when the local distributions are non-IID.

## 5.3. DCFM-B: Learning Globally Conditionally Independent Student.

One problem with DCFM-A is that, after a round of training, we obtain a new set of models $\{\mathbf{v}_\theta^{'(1)}, \mathbf{v}_\theta^{'(2)}, \ldots, \mathbf{v}_\theta^{'(k)}\}$. However, each updated model learns conditional independence with respect to the frozen models $\{\mathbf{v}_\theta^{(b)}\}_{b \neq a}$ that are obtained as a result of stage I, rather than the updated set $\{\mathbf{v}_\theta^{'(b)}\}_{b \neq a}$. This causes a potential mismatch between the models, solving which requires repeating the DCFM-A procedure several times until convergence. Although DCFM-A achieves convergence in a single iteration over the simple datasets shown in Fig. 2, this is not the case in more practical domains, such as images. Further, repeated training stages add to the computation and communication expense of training this group of local models.

We observe that, unlike discriminative models $F : \mathcal{X} \to \mathcal{Y}$ with intractable input space $\mathcal{X}$, the input space $\mathcal{Z}$ of a generative model $G$ is tractable for every decentralized

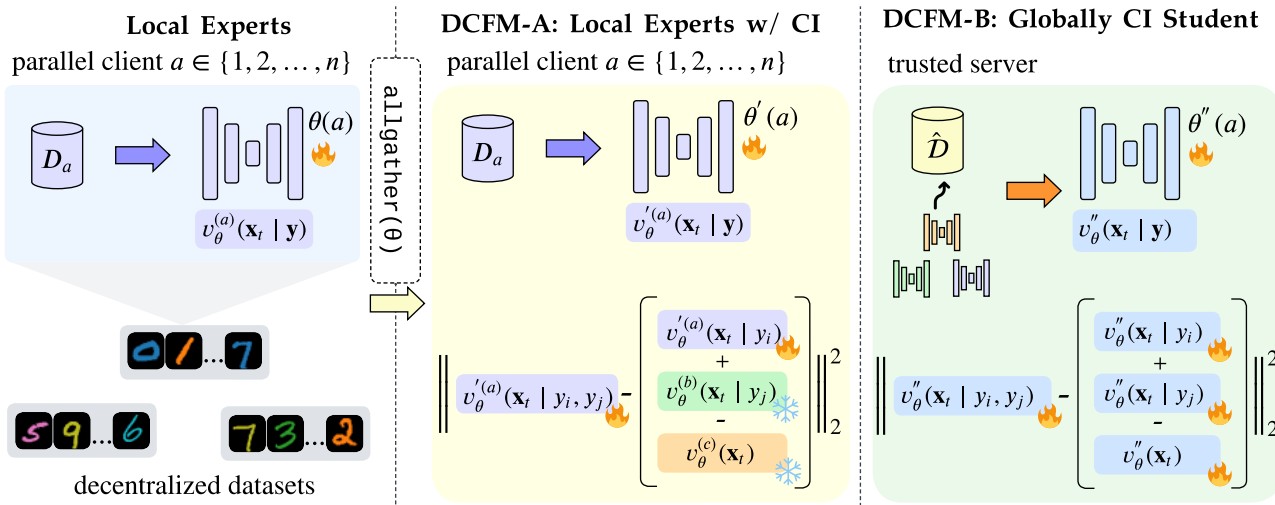

*Figure 4.* An overview of DCFM with 🔥 and ❄ indicating trainable and frozen parameters respectively. Stage 1 (left) trains local experts on local data. DCFM-A (middle) trains local experts with with cross-peer conditional independence constraints. DCFM-B (right) learns a globally conditionally independent monolithic student model.

client. Previous works, such as rectified flows (Liu et al., 2023) demonstrate that it is possible to optimize a flow model on its self-generated samples. We thus consider distilling a group of local experts into a single student. This distillation can be done on a trusted server node, or even any single client node after a single model exchange operation.

We define a peer matching objective as follows:

$$\mathcal{L}_{\text{student}}(\theta) = \mathop{\mathbb{E}}_{t,r,\hat{\mathbf{x}}_t} \left[ \|\mathbf{v}_\theta(\hat{\mathbf{x}}_t, t, \mathbf{y} \odot \mathbf{m}) - \bar{\mathbf{v}}_\theta^{(r)}(\hat{\mathbf{x}}_t, \mathbf{y})\|^2 \right] \tag{15}$$

where $\hat{\mathbf{x}}_t = \text{ODEInt}(\mathbf{x}_0, \bar{\mathbf{v}}_\theta^{(r)}, \mathbf{y}, 0, t)$ and $\bar{\mathbf{v}}$ is a frozen teacher. So, if $\hat{\mathbf{x}}_t$ belongs to the probability path of model $\bar{\mathbf{v}}_\theta^{(r)}$, we treat its response at $\hat{\mathbf{x}}_t$ as the ground truth velocity.

Distilling peer velocities following (15) allows us to generate samples that cover the union of $\mathcal{D}$. But, it does not sufficiently enable the recovery of missing combinations of $\mathbf{y}$. So, we again enforce conditional independence constraints between attributes during the peer distillation process. We add the following penalty to the training objective:

$$\mathbf{z}_\theta(\hat{\mathbf{x}}_t) = \left( \mathbf{v}_\theta(\hat{\mathbf{x}}_t) + \sum_{i=1}^{k} (\mathbf{v}_\theta(\hat{\mathbf{x}}_t, y_i) - \mathbf{v}_\theta(\hat{\mathbf{x}}_t)) \right)$$

$$\mathcal{L}_{\text{studentCI}}(\theta) = \mathbb{E}_{t,\mathbf{y},\hat{\mathbf{x}}_t} \left[ \|\mathbf{v}_\theta(\hat{\mathbf{x}}_t, \mathbf{y}) - \mathbf{z}_\theta(\hat{\mathbf{x}}_t)\|_2^2 \right] \tag{16}$$

Note that, we do not use any frozen model in (16), unlike (12). While learning the collective velocity fields of each peer, we also want to learn global marginals that are conditionally independent. The overall DCFM-B objective is:

$$\mathcal{L}_{\text{DCFM-B}}(\theta) = \mathcal{L}_{\text{student}}(\theta) + \lambda \mathcal{L}_{\text{studentCI}}(\theta) \tag{17}$$

As seen in Fig. 3 (bottom), instead of simply aggregating non-IID experts into a single model, DCFM-B also successfully explores their implied unobserved combination.

## 6. Experiments

DCFM is designed to train generative models within isolated data silos while enabling compositional generalization to novel attribute combinations. We evaluate DCFM based on two primary objectives: (1) **Compositional Generalization:** Assessing the model's ability to generate valid samples for attribute combinations unobserved during decentralized training, and (2) **Variant Trade-offs:** Characterizing the performance of the two DCFM variants on both known and novel compositions. To demonstrate the framework's versatility, we consider three distinct benchmarks: image composition via Colored MNIST (§ 6.1), medical imaging on Chest X-rays (§ 6.3), and spatial composition for unconstrained offline robotic planning (§ 6.2).

### 6.1. Colored MNIST

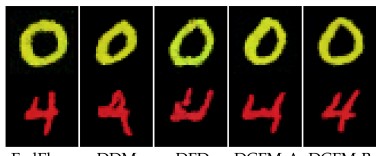

*Figure 5.* Novel compositions generated by DCFM vs. baselines; the exact combinations (0, yellow) and (4, red) do not exist in the training data of any client.

**Experiment Setting.** The Colored MNIST dataset (LeCun, 1998) consists of images $\mathbf{x} \in \mathbb{R}^{28 \times 28 \times 3}$ with corresponding $k = 2$ discrete attribute spaces, $\mathcal{Y}_1$ and $\mathcal{Y}_2$, which represent a set of ten numbers and ten colors respectively. The total attribute space, $\mathcal{Y}$ has $|\mathcal{Y}_1||\mathcal{Y}_2| = 100$ combinations. We consider these images to be distributed across $n = 10$ decentralized datasets, $\mathcal{D} = \{D_a\}_{a=1}^n$ with a total coverage of $C = 1/2$; that is, half of $\mathcal{Y}$ remains unobserved. Fur-

ther, we consider both IID and non-IID partition setups over $\mathcal{D}$. The non-IID partition utilizes the Dirichlet distribution Dir($\alpha = 0.1$).

**Metrics.** We utilize FID (Heusel et al., 2017), Precision (P), and Recall (R) (Kynkäänniemi et al., 2019) to measure the performance of $G_\theta$. We further disentangle these metrics in terms of *known* (observed across all clients in $\mathcal{D}$) and *novel* (denoted by superscripts $o$ and $*$ respectively).

*Table 1.* Performance on MNIST under IID and Non-IID settings.

| Method | IID | | Non-IID | |
|---|---|---|---|---|
| | FID$^o$ ↓ | FID$^*$ ↓ | FID$^o$ ↓ | FID$^*$ ↓ |
| FedFlow | 9.41 | 20.83 | 15.02 | 20.02 |
| FedFlow+L | 12.37 | 18.65 | 15.81 | 17.12 |
| DDM | **8.19** | 25.19 | 7.46 | 17.98 |
| DDM+L | 9.03 | 16.38 | 7.99 | 18.84 |
| DFD | 8.81 | 29.71 | **7.02** | 38.27 |
| DFD+L | 9.58 | 19.13 | 8.17 | 31.36 |
| DCFM-A (Ours) | 8.53 | **11.41** | 7.33 | 9.29 |
| DCFM-B (Ours) | 9.32 | 12.24 | 8.49 | **9.15** |

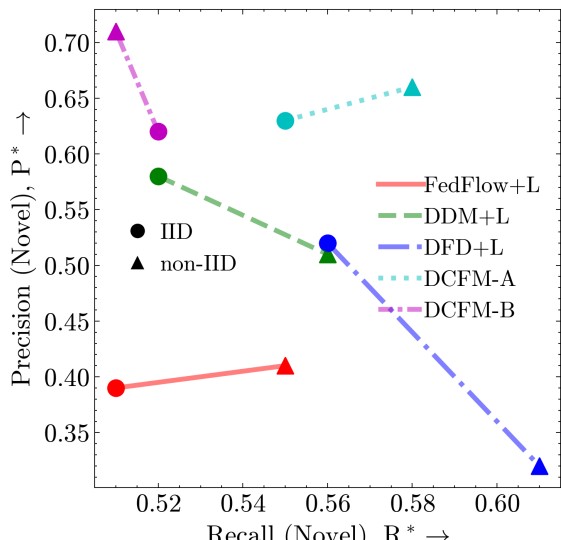

*Figure 6.* Precision vs. Recall on novel Colored MNIST compositions.

**Results and Analysis.** In Tab. 1, we compare DCFM with other decentralized approaches, where a +L indicates a model trained with a *local* CI penalty Eq. (10). We find that DFD (Hahn & Lee, 2025) shows the best result in recovering the known data, particularly under challenging non-IID conditions. However, they show poor generalization under novel attribute compositions, demonstrating a strong tendency to steer the samples towards known data regions. In contrast, both DCFM-A and DCFM-B show good performance in terms of FID$^o$ and FID$^*$, thus narrowing the gap between known and novel. We show some qualitative examples of novel combination generation in Fig. 5.

Fig. 6 shows the novel Precision (P$^*$) vs. novel Recall (R$^*$), using combinations that are unobserved in the decentralized $\mathcal{D}$. We observe that DCFM shows a notably improved P$^*$ compared to baselines, suggesting it is better able to capture the important components of the missing data. We find that DCFM-B shows a reduced recall score compared to DCFM-A, which indicates some loss in diversity due to learning from synthetic data.

### 6.2. Unconstrained Offline Robotic Planning

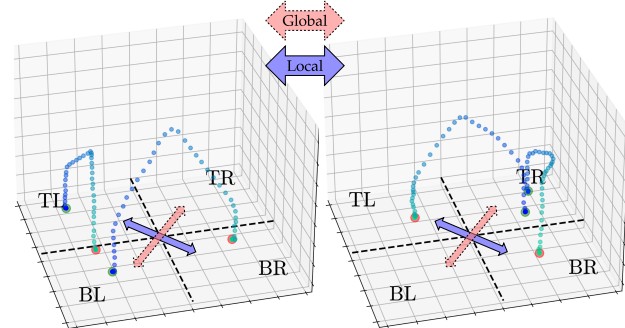

*Figure 7.* Two decentralized non-IID nodes, which observe movement of a cube between different quadrants. While each local expert can locally generalize to TL↔BR movement, they need to learn global composition to achieve TR↔BL movement.

**Experiment Setting.** The `cube-single-play` dataset from the OGBench benchmark (Park et al., 2025) contains trajectory data of a robot arm that moves a cube around within a confined 3D space. We characterize a complete trajectory plan by $k = 2$ attributes (Src, Dest), which indicate the location where the cube is picked up, and the destination where it is dropped off by the robot arm respectively. To make the attributes categorical, we divide the horizontal $XY$ plane into 4 quadrants. We thus have two attributes, $\mathcal{Y}_1, \mathcal{Y}_2 \in \{$TL, BL, BR, TR$\}$, and the total attribute space $\mathcal{Y}$ has a cardinality of $|\mathcal{Y}_1||\mathcal{Y}_2| = 16$.

*Table 2.* Success rate (SR) of DCFM vs. baselines on `cube-single-play`, averaged over 3 groups.

| Method | Partition | SR$^o$ | SR$^*$ |
|---|---|---|---|
| FedFlow+L | IID | **58.33 ± 2.62** | 39.67 ± 4.11 |
| (Tun et al., 2023) | Non-IID | 27.67 ± 7.76 | 9.0 ± 3.74 |
| DDM+L | IID | 53.33 ± 2.05 | 38.33 ± 5.73 |
| (McAllister et al., 2025) | Non-IID | 67.67 ± 2.87 | 29.67 ± 2.87 |
| DFD+L | IID | 58.0 ± 2.16 | 40.0 ± 4.9 |
| (Hahn & Lee, 2025) | Non-IID | **68.67 ± 1.25** | 18.33 ± 5.44 |
| DCFM-A | IID | 57.67 ± 2.62 | **56.33 ± 5.31** |
| (Ours) | Non-IID | 68.33 ± 2.87 | 53.0 ± 2.94 |
| DCFM-B | IID | 54.0 ± 3.74 | 53.33 ± 2.36 |
| (Ours) | Non-IID | 65.67 ± 2.49 | **54.67 ± 2.49** |

Following (Janner et al., 2022), we define a trajectory $\mathbf{x}$ as a tuple $[\mathbf{s}, \mathbf{a}]$, where $\mathbf{s} = (\mathbf{s}^0, \mathbf{s}^1, \ldots, \mathbf{s}^{H-1})$ is a sequence of *observations*, $\mathbf{a} = (\mathbf{a}^0, \mathbf{a}^1, \ldots, \mathbf{a}^{H-1})$ is a corresponding

sequence of *actions*, and $H$ is the planning horizon. We want to learn a planning model, $\mathbf{v}_\theta([\mathbf{s}_t, \mathbf{a}_t], t, \mathbf{y})$, where $\mathbf{s}_t$ and $\mathbf{a}_t$ indicate a noisy trajectory at timestep $t$.

The trajectory data is split across $n = 2$ decentralized datasets $\mathcal{D} = \{D_1, D_2\}$ for both IID and non-IID partitions. We set the total coverage to $3/4$, omitting trajectory data with the four attribute combinations (TL, BR), (TR, BL), (BR, TL), and (BL, TR) respectively. That is, the robot may have data of moving between horizontal quadrants (like TL↔TR) or vertical quadrants (TL↔BL). But it has no observed information about moving between diagonal quadrants, like (BL↔TR). In order to learn diagonal movement, the robot planner needs to understand that *the destination of a trajectory is independent of its source*. We provide a visual overview of the problem in Fig. 7.

**Goal.** We rely on the planner to suggest a feasible goal. Given a condition vector $\mathbf{y}$, the flow planner $\mathbf{v}_\theta$ generates a trajectory $\mathbf{x} = [\mathbf{s}, \mathbf{a}]$. We take the first and final observations within a trajectory, $\mathbf{s}_0$ and $\mathbf{s}_{H-1}$, and extract the cube positions $c^0$ and $c^{H-1}$, respectively. We consider a plan *feasible* if: (i) the cube is on the floor at both $c^0$ and $c^{H-1}$, (ii) $c^0$ is located at the given source quadrant $y_1$, and (iii) $c^{H-1}$ is at the given destination quadrant $y_2$. Only if the initial plan defines a feasible goal $c^{H-1}$, we execute the plan with the inpainting-based method proposed by Janner et al. (2022).

**Evaluation Metrics.** We report Success Rate (SR), defined as the percentage of trials where: (a) the generated plan's start and end states match the queried quadrants, and (b) the executed trajectory places the cube within a threshold distance of the target.

**Results and Analysis.** From Tab. 2, we observe that DCFM significantly outperforms all baselines in planning under novel attribute compositions, while retaining comparable performance for the compositions known in the dataset.

### 6.3. Disease Co-Occurrence in Chest X-Rays

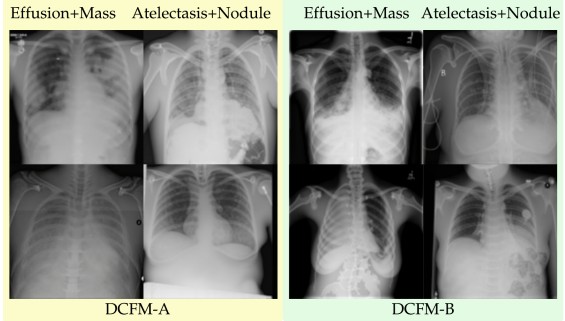

*Figure 8.* Disease combinations generated by DCFM.

**Experiment Setting.** The NIH Chest X-ray 14 dataset (Wang et al., 2017) contains over 100K chest X-ray (CXR) images labeled with 14 disease attributes. We use images of

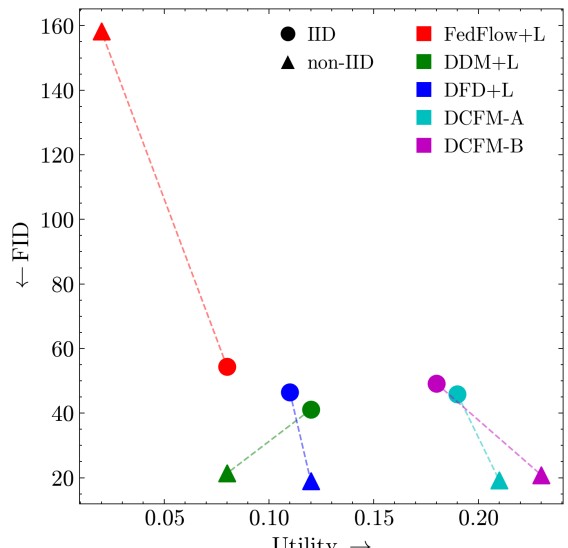

*Figure 9.* FID vs. Utility on NIH Chest X-ray 14.

size $256 \times 256$, that are distributed across $n = 4$ decentralized datasets $\mathcal{D}$ with IID and non-IID partitions.

**Attribute Sparsity.** Given $k = 14$ binary attributes, $|\mathcal{Y}|$ grows steeply to $2^{14}$. However, a large portion of the product space is infeasible. For example, it is impossible to see all 14 diseases occur simultaneously, and at most 9 diseases ever co-occur together in the training data (with a count of 2). We find that $\sim$99% of the training data contains 3 or fewer co-occurring diseases. Instead of using the total attribute space $\mathcal{Y}$, we define a *feasible attribute space* $\mathcal{Y}_\mathcal{F}$ as the set of all $\mathbf{y} \in \mathcal{Y}$ with a frequency of at least 0.1% in $\mathcal{D}$. This leaves us with a feasible space $\mathcal{Y}_\mathcal{F}$ with 54 combinations. **Goal.** Our objective is to correctly synthesize the disease combinations in $\mathcal{Y}_\mathcal{F}$, by learning a decentralized model that disentangles the conditional dependencies between diseases.

**Evaluation Metrics.** We use FID to measure the generative performance of the trained decentralized flows. We do not separate the FID into known vs. novel, as we lack sufficient ground truth data for the rare combinations in the dataset. To quantify the usefulness of the synthetic samples, we measure **utility (U)**, as the *compositional* recall (sensitivity) score on the real test set of a downstream classifier trained on purely synthetic data. Suppose $\mathcal{A}$ is a set of co-occurring diseases. We define a compositional recall as,

$$R(\mathcal{A}) = \frac{\{\mathbf{y}_{\text{pred}} = \mathbb{1}_\mathcal{A}\} \wedge \{\mathbf{y}_{\text{true}} = \mathbb{1}_\mathcal{A}\}}{\text{count}(\mathbf{y}_{\text{true}} = \mathbb{1}_\mathcal{A})}, \quad (18)$$

where $\mathbb{1}_\mathcal{A} \in \{0,1\}^k$ is a binary attribute vector that is 1 at the indices $\mathcal{A}$. We define U as the macro-average of Eq. (18) over the distinct combinations in $\mathcal{Y}_\mathcal{F}$.

**Models.** Across all baselines, we train a latent flow matching model (Rombach et al., 2022; Esser et al., 2024) at

$256 \times 256$ resolution, utilizing the latent space of a pretrained autoencoder. We use a ResNet-18 model as the downstream classifier.

**Results and Analysis.** From Fig. 9, we observe that, though DCFM achieves similar quality as other decentralized generative approaches, a classifier trained on DCFM shows improved sensitivity to the joint occurrence of chest diseases. This suggests that DCFM is able to reduce the entanglement between some diseases due to training data correlations, thus synthesizing more representative features when generating rare combinations. Fig. 8 shows some qualitative examples of images generated by DCFM.

### 6.4. Analysis of Communication and Computation Cost

**Communication Cost.** Suppose sending or receiving the total parameters in the model imply *1 unit of communication cost*. Further, for DDM and DFD, we assume a particular client receives all expert models for mixture inference.

| Method | Cost Per Client |
|--------|----------------|
| FL | $2T$ |
| DDM | $N-1$ |
| DFD | $N-1$ |
| DCFM | $RN$ |
| DCFM-B | $2$ |

*Table 3.* Communication cost (send/receive $\boldsymbol{\theta}$) per client.

We show the cost in Tab. 3, where $T$ is the number of federated training rounds, $R$ is the number of DCFM-A rounds, $N$ is the number of clients. Diffusion or flow matching training usually consists of very large number of federated rounds, thus requiring a large number of federated rounds ($T$) (e.g. $T \geq 100$; Tun et al. (2023) use up to 300 rounds and 5 epochs per round). In contrast, DDM doesn't require any training time communication; only a chosen client receives $N-1$ other models at the end of local training. DFD (Hahn & Lee, 2025) does not actually require any model transmission for inference. However, this makes the communication cost potentially unbounded, as it transmits the diffusion model output per timestep, per generated sample. We thus treat DFD like DDM, and perform centralized inference from a single client. DCFM-A generally requires 1-3 rounds ($R \sim 2$ on average). DCFM-B has a constant communication cost of 2, as every peer uploads local model to central server, and receives a distilled model in return.

**Computation.** Given the variability in GPU hours or clock times, we approach computational costs in terms of effective forward/backward passes required during training. We group the methods having similar cost, and provide an average of their FID-novel metric (FID) from Tab. 1 (averaged over IID vs Non-IID, and over methods) in Tab. 4

*Table 4.* Cost per local training step of various methods vs. average FID* (from Tab. 1), with batch size $B$ and #P parallel nodes.

| Method | #P | Fwd. | Bwd. | Avg. FID* |
|--------|----|------|------|-----------|
| (Baselines) FL, DDM, DFD | n | $1 \times B$ | $1 \times B$ | 25.33 |
| Baselines+Local CI | n | $\sim 4 \times B$ | $\sim 4 \times B$ | 20.24 |
| DCFM-A | n | $\sim 4 \times B$ | $\sim 2 \times B$ | 10.35 |
| DCFM-B | 1 | $\sim 5 \times B$ | $\sim 4 \times B$ | 10.69 |

Since enforcing Eq. (10) depends on $k$, a large number of attributes would incur a significant cost. In practice, we take an expectation over pairs of attributes, which requires computing 4 distinct terms in Eq. (10). In Tab. 4, we thus write the cost of a forward / backward pass as $4B$, when using local conditional independence. We also observe that, although DCFM requires additional computation over classical approaches, it is able to utilize the extra compute for improved compositional performance. Even if baseline methods are provided extra compute, they cannot sufficiently capture unobserved combinations as they do not explicitly enforce global compositional structure.

**Performance.** From observing Tab. 1, Tab. 2, and Fig. 6, we consistently find that DCFM-B underperforms w.r.t. DCFM-A on the *known* attributes. Further, Fig. 6 shows that DCFM-B has reduced recall, suggesting reduced diversity of the samples. We hypothesize this to be caused by the synthetic distillation process, as the experts obtained from Stage I are not necessarily perfect approximators of the real data. Nevertheless, DCFM-B is able to achieve comparable or better performance on the unobserved or novel attributes while requiring less computation and communication.

## 7. Concluding Remarks

We introduced Decentralized Compositional Flow Matching (DCFM), a framework that enables generative models to achieve compositional generalization across isolated data silos without exchanging raw data. Unlike prior decentralized approaches, which often suffer from factor entanglement or incompatible latent spaces, DCFM structurally enforces conditional independence through peer-to-peer knowledge distillation in a shared flow velocity space. This design allows novel attribute combinations to emerge from decentralized sources, even when no individual silo possesses the information required to support such compositions.

Across diverse benchmarks, including Colored MNIST, chest X-rays, and robotic spatial planning, DCFM significantly outperforms federated learning and mixture-of-experts baselines in recovering unobserved or rare combinations. This work establishes a foundation for scaling generative systems in locality-sensitive and resource-constrained environments, where the ability to synthesize global knowledge from fragmented, localized observations is essential.

**Acknowledgements:** This work was supported by the National Science Foundation (award #2500983). Any opinions, findings, and conclusions or recommendations expressed in this material are those of the authors and do not necessarily reflect the views of NSF.

## Impact Statement

We can characterize the present state of machine learning as follows: the model architectures and training algorithms are known; the model weights may even publicly available to download (for instance, Stable Diffusion or FLUX models). However, the data remains increasingly private. With open architectures and training methods, curated data often ends up as the deciding factor in the performance of generative models. It is likely that people will be significantly less inclined to share their private, local data in the near future, as data would become something akin to an economic *asset*. Under these circumstances, we believe our paper raises an important line of investigation: *how should we learn good decentralized generative models, that do not violate the locality of data?*

We also envision a promising avenue of research at the intersection of compositional generalization and decentralized learning. Although they initially appear to be quite different problems, we find that they are also structurally similar, in the sense that both aim to combine 'components' to learn a larger 'whole'. Our work takes an initial step toward bridging these domains, demonstrating that generative learning from isolated data fragments can yield a much broader synthesis than simply the union of components.

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

## A. Compositional Velocity Fields

In order to justify the factorization of the joint conditional velocity field in Eq. (5), we first derive the the compositional score function in § A.1. In § A.2, we then show how the velocity factorization in Eq. (5) naturally arises from the compositional score, by using the relationship between the score function and flow velocities.

### A.1. Deriving the Compositional Score Function

Let $\{y_1, y_2, \ldots, y_k\}$ be a set of condition variables associated with data $\mathbf{x}$. The joint conditional score of $\mathbf{x}$ is then

$$\nabla \log p_t(\mathbf{x} \mid y_1, y_2, \ldots, y_k) = \nabla \log \frac{p_t(\mathbf{x}) p_t(y_1, y_2, \ldots, y_k \mid \mathbf{x})}{p_t(y_1, y_2, \ldots, y_k)} \tag{19}$$

$$= \nabla \log p_t(\mathbf{x}) + \nabla \log \underbrace{p_t(y_1, y_2, \ldots, y_k \mid \mathbf{x})}_{\text{Joint posterior}}. \tag{20}$$

By assuming $\{y_i\}_{i=1}^k$ are *conditionally independent*, we can utilize Eq. (1) to decompose the joint posterior,

$$\nabla \log p_t(\mathbf{x} \mid y_1, y_2, \ldots, y_k) = \nabla \log p_t(\mathbf{x}) + \nabla \log \left( \underbrace{\prod_{i=1}^k p_t(y_i \mid \mathbf{x})}_{\text{by Eq. (1)}} \right) \tag{21}$$

$$= \nabla \log p_t(\mathbf{x}) + \sum_{i=1}^k \nabla \log p_t(y_i \mid \mathbf{x}) \tag{22}$$

The factorization in Eq. (22) enables compositional score sampling with classifier-free guidance (Ho & Salimans, 2022).

Let $w_i$ be a temperature parameter associated with the condition $y_i$, such that,

$$\nabla \log \hat{p}_t(\mathbf{x} \mid y_1, y_2, \ldots, y_k) = \nabla \log p_t(\mathbf{x}) + \sum_{i=1}^k w_i \nabla \log p_t(y_i \mid \mathbf{x}) \tag{23}$$

$$= \nabla \log p_t(\mathbf{x}) + \sum_{i=1}^k w_i \left( \nabla \log p_t(\mathbf{x} \mid y_i) - \nabla \log p_t(\mathbf{x}) \right), \tag{24}$$

where $\hat{p}_t(\mathbf{x} \mid y_1, y_2, \ldots, y_k) \propto p_t(\mathbf{x})^{(1 - \sum_i w_i)} \prod_{i=1}^k p_t(\mathbf{x} \mid y_i)^{w_i}$ is a geometric average of the unconditional likelihood and the $k$ marginal conditional likelihoods.

### A.2. Deriving and Justifying the Compositional Velocity Field

Eq. (24) has been utilized previously by several existing works (Liu et al., 2022; Ajay et al., 2023; Gaudi et al., 2025), and directly admits Eq. (4) for diffusion models. However, it is not well studied whether this compositional procedure extends to flow velocities $\mathbf{v}$ as well. We provide a brief validation in the following.

**Proposition.** *The compositional flow velocity, $\mathbf{v}_t(\mathbf{x}) + \sum_{i=1}^k w_i \left( \mathbf{v}_t(\mathbf{x} \mid y_i) - \mathbf{v}_t(\mathbf{x}) \right)$ is consistent with the compositional score, $\nabla \log p_t(\mathbf{x}) + \sum_{i=1}^k w_i \left( \nabla \log p_t(\mathbf{x} \mid y_i) - \nabla \log p_t(\mathbf{x}) \right)$.*

*Proof.* We first establish certain assumptions over the probability path $p_t$ associated with the velocity field $\mathbf{v}_t$. Given the source distribution $p_0 \sim \mathcal{N}(\mathbf{0}, \mathbf{I})$ and the data distribution $p_1$, we may define $p_t$ as

$$p_t(\mathbf{x} \mid \mathbf{x}_1) = \mathcal{N}(\mathbf{x} \mid \alpha_t \mathbf{x}_1, \sigma_t^2 \mathbf{I}), \tag{25}$$

where $\mathbf{x}_1 \sim p_1$, and the pair $(\alpha_t, \sigma_t)$ follow the boundary conditions $(\alpha_0, \sigma_0) = (0, 1)$, $(\alpha_1, \sigma_1) = (1, 0)$ such that $p_t$ results in the noise and data distribution at $t = 0$ and $t = 1$ respectively.

Next, we consider a conditional velocity field $\mathbf{v}_t(\mathbf{x} \mid y)$ associated with the path $p_t$.

By *Lemma 1* from Zheng et al. (2023), if $p_t$ follows Eq. (25), the velocity $\mathbf{v}_t$ can be related to the score function $\nabla \log p_t(\mathbf{x} \mid y)$ by

$$\mathbf{v}_t(\mathbf{x} \mid y) = a_t \mathbf{x} + b_t \nabla \log p_t(\mathbf{x} \mid y), \tag{26}$$

where,

$$a_t = \frac{\dot{\alpha}_t}{\alpha_t}, \quad b_t = (\dot{\alpha}_t \sigma_t - \alpha_t \dot{\sigma}_t)\frac{\sigma_t}{\alpha_t}. \tag{27}$$

We can rearrange Eq. (26) to express the score in terms of the velocity as follows,

$$\nabla \log p_t(\mathbf{x} \mid y) = \frac{\mathbf{v}_t(\mathbf{x} \mid y) - a_t \mathbf{x}}{b_t}. \tag{28}$$

Then Eq. (28) can be used in the RHS of Eq. (24),

$$\nabla \log \hat{p}_t(\mathbf{x} \mid y_1, y_2, \ldots, y_k) = \nabla \log p_t(\mathbf{x}) + \sum_{i=1}^{k} w_i \left( \nabla \log p_t(\mathbf{x} \mid y_i) - \nabla \log p_t(\mathbf{x}) \right)$$

$$= \frac{\mathbf{v}_t(\mathbf{x}) - a_t \mathbf{x}}{b_t} + \sum_{i=1}^{k} w_i \left( \frac{\mathbf{v}_t(\mathbf{x} \mid y_i) - a_t \mathbf{x}}{b_t} - \frac{\mathbf{v}_t(\mathbf{x}) - a_t \mathbf{x}}{b_t} \right) \tag{29}$$

$$= \frac{\mathbf{v}_t(\mathbf{x}) - a_t \mathbf{x}}{b_t} + \frac{1}{b_t} \sum_{i=1}^{k} w_i \left( \mathbf{v}_t(\mathbf{x} \mid y_i) - \mathbf{v}_t(\mathbf{x}) \right) \tag{30}$$

$$= \frac{1}{b_t} \left[ \mathbf{v}_t(\mathbf{x}) + \sum_{i=1}^{k} w_i \left( \mathbf{v}_t(\mathbf{x} \mid y_i) - \mathbf{v}_t(\mathbf{x}) \right) - a_t \mathbf{x} \right] \tag{31}$$

By re-arranging the terms we get

$$a_t \mathbf{x} + b_t \nabla \log \hat{p}_t(\mathbf{x} \mid y_1, y_2, \ldots, y_k) = \mathbf{v}_t(\mathbf{x}) + \sum_{i=1}^{k} w_i \left( \mathbf{v}_t(\mathbf{x} \mid y_i) - \mathbf{v}_t(\mathbf{x}) \right) \tag{32}$$

$$\hat{\mathbf{v}}_t(\mathbf{x} \mid y_1, y_2, \ldots, y_k) = \mathbf{v}_t(\mathbf{x}) + \sum_{i=1}^{k} w_i \left( \mathbf{v}_t(\mathbf{x} \mid y_i) - \mathbf{v}_t(\mathbf{x}) \right) \tag{33}$$

Thus, we show that the compositional velocity $\hat{\mathbf{v}}_t(\mathbf{x} \mid y_1, y_2, \ldots, y_k)$ defined in Eq. (5) is consistent with the compositional score $\nabla \log \hat{p}_t(\mathbf{x} \mid y_1, y_2, \ldots, y_k)$ (Eq. (24)).

$\square$

