# OpenReview forum: "Compositional Generative Modeling from Decentralized Data"
_ICML.cc/2026/Conference — ICML 2026 regular_

### Official Review · Reviewer_ToUV · 2026-02-14

**Soundness:** 2
**Presentation:** 3
**Significance:** 3
**Originality:** 3
**Overall Recommendation:** 5
**Confidence:** 2

**Summary:**

This paper studies conditional generative modeling under decentralized data, focusing on compositional generalization: generating samples for attribute combinations that are unseen globally. The authors argue that existing methods (federated averaging and mixture-/routing-based methods) fail due to two key issues: (i) lack of conditional-independence (CI) structure, and (ii) incompatibility of learned vector fields across local experts, especially under non-IID cases. The paper proposes Decentralized Compositional Flow Matching (DCFM). It introduces a vector field built from marginal-conditioned vector fields and adds a CI penalty so the joint-conditioned field matches this compositional surrogate. Two variants are presented: DCFM-A aligns multiple local experts via cross-peer compositional consistency, while DCFM-B is more efficient by distilling experts into one student with a global compositional constraint. Generally, I found the paper well-written, with clear structure and strong results.

**Compliance With Llm Reviewing Policy:**

Affirmed.

**Final Justification:**

All of my concerns have been addressed

**Key Questions For Authors:**

- Do you have a quantitative metric showing experts become more “compatible” after cross-peer training?
- How does performance scale with increasing heterogeneity?
- Do you plan to make the code publicly available?

**Limitations:**

See Weakness.

**Strengths And Weaknesses:**

Strength:
- Well-motivated problem
- Clear diagnosis of failure modes. The paper finds two causes of failure: structural (CI/compositionality) and optimization/representation (cross-expert incompatibility), supported with toy experimental results.
- Strong experimental result in MNIST image generation, robotic planning and medical imaging.

Weakness:
-Details affecting reproducibility are not provided.
- Ablation study is missing. The paper would be stronger with ablations showing sensitivity.

---

> ### Author Rebuttal · Authors · 2026-03-31
>
> We thank the reviewer for their positive feedback. We are encouraged that the reviewer finds our problem to be *well-motivated* (S1), our elucidation of the failure modes of existing paradigms to be *clear and empirically supported* (S2), and our experimental results *strong* (S3).
>
> ## Response to Weaknesses & Questions
>
> **W1. Training details:** Please refer to our response to reviewer UAfn, where we provide additional details.
>
> **W2. Ablations:** Please refer to **Table R2** (resp. to swmX), **Table R3** (below), **Table R5, R6** (resp. to jQNd) for ablations.
>
> **Q1.** **Quantitative metric regarding “compatibility”:** This is an interesting question. While we do not present any metric in the paper, we posit that it is possible to design a metric based on the conditional independence assumption (Eqn. 1).
>
> We first clarify what we mean by ***compositional compatibility***. Existing mixture approaches, like DDM and DFD, use a weighted mixture of diffusion or flow matching models, of the general form: $\tilde{v}\_{\theta} = \sum\_a r_a v_{\theta}^{(a)}$ , where, $v\_\theta^{(a)}$ is the model local to the $a$-th client/node, and $r\_a$ is the routing weight given to that particular client. Even if each individual $v\_\theta^{(a)}$ is compositional, the collective $\tilde{v}\_\theta$ are not necessarily so.
>
> Now, let $\tilde{v}\_\theta(x_t \mid y_1, \dots, y_k) = \sum\_a r\_a v\_\theta^{(a)}(x\_t \mid y\_1, \dots, y\_k)$ represent the mixture of the *joint* velocities. Similarly, we can define:
>
> - mixtures of *marginal* velocities $\tilde{v}_\theta(x_t \mid y_i)$ for each marginal attribute $y_i$
> - a mixture of *unconditional* velocities $\tilde{v}_\theta(x_t \mid \emptyset)$.
>
> If the mixture of flow models are *compatible* in terms of composition, they should satisfy Eqn 1:
>
> $p(y_1, \dots, y_k \mid x) = \prod_{i=1}^k p(y_i \mid x)$
>
> Next, from the mixture of models $\tilde{v}\_\theta(x \mid y)$, we can approximate the implicit mixed classifier, $\tilde{p}\_\theta (y \mid x)$, following Li et al. 2023 [a].
>
> Subsequently, to define a metric, we can measure,
>
> $L\_{cc} = \mathbb{E}\_{(x, y)\in\mathcal{D}}\left[D\_{JS}\left( \tilde{p}\_\theta(y\_1, \dots, y\_k \mid x) || \prod\_i\tilde{p}\_\theta(y\_i \mid x)\right)\right]$
>
> Where $D_{JS}$ is the Jensen-Shannon divergence. The above metric has been previously used by Gaudi et al. 2025, for measuring conditional independence relationships in a single diffusion model. Simply put, $L_{cc}$ measures if the inference-time combination of a group of diffusion or flow matching models are sufficiently conditional independent (lower is better).
>
> **TableRcc** below measures $L_{cc}$ for our method vs. baselines on ColoredMNIST (non-IID). We find that the $L_{cc}$ score for DCFM-A is much lower than that of the baselines, suggesting that the group of models are more compositionally compatible. (We exclude DCFM-B from this comparison, as it is a single distilled model instead of a group of models.)
>
> | Method | $L_{cc} (\downarrow)$ |
> | --- | --- |
> | DDM | 3.21 |
> | DDM+L | 2.38 |
> | DCFM-A | 1.46 |
>
> **(Q2)** **Scaling w.r.t. heterogeneity.** We perform a small ablation w.r.t. heterogeneity, by fixing the number of clients to $n=4$, and varying the parameter of the Dirichlet partitioning from $\alpha=0.1 \dots 5$, where increasing $\alpha$ makes the distribution more homogenous.
>
> **Table R3** measures the FID-novel (FID$^*$) metric for colored MNIST across three different $\alpha$ values:
>
> | Method | $\alpha=0.1$ | $\alpha=1$ | $\alpha=5$ |
> | --- | --- | --- | --- |
> | DDM+L | 17.98 | 19.02 | 23.61 |
> | DCFM-A | 9.29 | 11.23 | 11.58 |
> | DCFM-B | 9.15 | 10.87 | 12.62 |
>
> We observe that for novel compositions, the performance of DCFM, as well as Decentralized Diffusion (DDM) baseline), decrease with higher $\alpha$ (more homogeneity), as local experts ideally prefer heterogeneity.
>
> **(Q3)** Yes, we will release code to reproduce the experimental results.
>
> [a] Li et al., “Your Diffusion Model is Secretly a Zero-Shot Classifier.”, CVPR 2023.
> [b] Gaudi et al., “CoInD: Enabling Logical Compositions in Diffusion Models”, ICLR 2025.

---

> > ### Author Rebuttal · Reviewer_ToUV · 2026-04-02
> >
> > I thank the authors for their response. All of my concerns have been addressed. I will keep the score.

---

> > > ### Author Response · Authors · 2026-04-03
> > >
> > > We thank the reviewer for acknowledging our rebuttal, and the general positive feedback. We will incorporate the discussion on quantifying compositional compatibility and other clarifications in the appendix of the revised manuscript.

---

### Official Review · Reviewer_jQNd · 2026-03-05

**Soundness:** 3
**Presentation:** 3
**Significance:** 3
**Originality:** 3
**Overall Recommendation:** 4
**Confidence:** 4

**Summary:**

This paper proposes a framework for training generative models from decentralized datasets called DCFM without sharing raw data. The approach enforces conditional independence across attributes and aligns locally trained flow models to enable compositional generalization to unseen attribute combinations. The authors introduce two variants: a MoE-style method and a distilled global student model. Experiments on three different datasets demonstrate improved performance compared to some baselines.

**Compliance With Llm Reviewing Policy:**

Affirmed.

**Key Questions For Authors:**

1) The paper compares DCFM-A and DCFM-B, but could the authors provide a more systematic ablation isolating the effect of key components (e.g., conditional independence penalty, flow matching alignment)? This would help clarify which parts contribute most to the gains.

2) How sensitive is the method when the conditional independence assumption is violated in practice? Additional discussion or experiments would help understand robustness.

3)  Could the authors comment on how the approach scales to larger generative models (e.g., higher-resolution image generation or diffusion models) in terms of computation and communication cost?

4)  What is the communication cost compared to standard federated learning baselines, and does performance remain stable under limited communication rounds

5) In real-world datasets, attributes are often correlated (e.g., in medical data). How does the method handle such correlations in practice?

6) Could the authors comment on the expected performance of the method on more complex or larger-scale generative datasets?

7) How sensitive is the method to highly heterogeneous client data distributions and potential misalignment between local experts?

8) Could the authors comment on the stability of training across decentralized clients and whether the approach requires careful hyperparameter tuning?

**Limitations:**

The paper mentions some technical limitations, but the discussion is brief. A more detailed discussion of practical limitations, robustness when assumptions do not hold, and potential real-world risks would strengthen the paper.

**Strengths And Weaknesses:**

Soundness: The paper proposes a technically sound framework for decentralized compositional generative modeling based on flow matching with conditional independence constraints. The method is well motivated, and the objectives are largely consistent with the problem formulation. Experiments across multiple datasets, including Colored MNIST, a chest X-ray dataset, and a robotic trajectory planning benchmark, support the main claims and show improved recovery of unseen attribute combinations compared to baselines. However, the theoretical justification remains mostly conceptual, and the framework relies on conditional independence assumptions that may not always hold in realistic settings. In addition, while two variants (DCFM-A and DCFM-B) are evaluated, a more systematic ablation study would help clarify the contribution of individual components. Some benchmarks, such as Colored MNIST, are relatively simple, and additional evaluation on larger-scale or more complex datasets could further strengthen the empirical validation.

Presentation: The paper is generally well organized, with clear motivation and helpful figures illustrating the compositional learning setup. However, some technical sections; particularly the formulation of the objectives; are dense, and several notations in the formulas are introduced without sufficient explanation, making parts of the derivations difficult to follow. A clearer high-level overview of the full pipeline and more careful introduction of the notation would improve readability.

Significance: The paper addresses an important problem of learning generative models from decentralized datasets with compositional structure, which is relevant for privacy-sensitive domains such as healthcare or distributed sensing. The experimental results are promising, although the broader impact of the method will depend on how well it scales to larger models and more complex real-world datasets.

Originality :The combination of flow matching with conditional independence constraints for decentralized compositional learning is interesting. While the individual components build on existing work in generative modeling and compositional learning, their integration to enable compositional generalization across decentralized data silos represents a meaningful contribution.

---

> ### Author Rebuttal · Authors · 2026-03-31
>
> We thank the reviewer for their positive feedback. We appreciate their recognition of our work as a technically sound, well-motivated, and empirically validated framework **(Soundness)**. They also highlighted the novelty of our contribution **(Originality)**, its significance for privacy-sensitive domains like healthcare and distributed sensing **(Significance)**, and the paper’s clear organization and helpful figures **(Presentation)**.
>
> ## Response to Weaknesses & Questions
>
> **(Soundness W1, Q2, Q5) Conditional independence assumptions.**
> We agree that real-world data may not necessarily be conditionally independent. However, (1) there are sufficiently many applications where we know the attributes are conditionally independent, and (2) it is still of interest to *assume* conditional independence for generative learning, given the goal of exploring novel compositional combinations. In our paper, we use conditional independence, i.e. $p(y\_1, y\_2 \mid x) = p(y\_1 \mid x)p(y\_2 \mid x)$, as a useful inductive bias that enables compositional generalization.
>
> **(Soundness W2, Q1)** **Ablation.** We provide 4 limited ablations:
>
> - **Table R2** (resp. to swmX) shows how increasing the number of clients affect performance.
> - **Table R3** (resp. to reviewer ToUV) shows how client heterogeneity affects performance.
> - **Table R5** (below) shows the effect of number of DCFM-A rounds, on ColoredMNIST (non-IID).
>
> | # rounds | FID$^o$ | FID$^*$ |
> | --- | --- | --- |
> | 1 | 7.57 | 13.82 |
> | 2 | 7.33 | 9.29 |
> | 3 | 7.91 | 9.03 |
> | 4 | 7.48 | 8.81 |
>
>  We find that, the biggest improvement in performance comes after 2 rounds, with subsequent rounds leading to slight performance gains.
>
> - **Table R6** (see Q8) shows effect of $\lambda$ (strength of CI penalty).
>
> **(Soundness W3, Q3, Q6) Larger scale or more complex datasets.**
> **While we agree with the reviewer that more complex datasets could improve both the utility and the validation of the paper, scaling multi-client decentralized learning for diffusion models remains a significant computational and data challenge. Contemporary works (such as our baseline DFD (Hahn & Lee, NeurIPS 2025)) only provide experiments at 32 x 32 resolution, and only for image generation (MNIST, CIFAR10, CelebA). In contrast, we provide experiments over more diverse domains, such as robotic trajectory planning and medical image generation.
>
> **(Presentation W1)** **Complexity of notation.** We thank the reviewer for the feedback; we will try our best to improve the clarity of our notation and presentation wherever possible.
>
> We believe the complexity of notation primarily stems from the fact that our paper incorporates both decentralized and compositional learning into generative methods. Existing works on flow matching pose learning objectives implicitly from the perspective of *a single local client*, and only consider a *single attribute variable*, $\mathcal{Y}=\mathcal{Y}\_1$. In our paper, we need to present objectives considering **(a) multiple clients/nodes**, and **(b) interactions between attribute combinations** in the product space $\mathcal{Y = Y\_1 \times \dots \times Y}\_k$.
>
> **Q4. Communication Cost Compared to FL.**
>
> Suppose the total parameters in the model imply 1 unit of communication cost. Then the communication cost of DCFM vs. federated learning is as follows:
>
> | Method | Cost per client |
> | --- | --- |
> | FL | 2T |
> | DCFM-A | RN |
> | DCFM-B | 2 |
>
> Where **T** is the number of federated training rounds, **R** is the number of DCFM-A rounds, **N** is the number of clients. Diffusion or flow matching training usually consists of very large number of federated rounds, thus requiring a large number of federated rounds (T) (e.g. T ≥ 100; Tun et al. 2023 use up to 300 rounds and 5 epochs per round). In contrast, DCFM-A generally requires 1-3 rounds (R ~ 2 on average). DCFM-B has a constant communication cost, as every peer uploads local model to central server, and receives a distilled model.
>
> **Q7. Sensitivity to Heterogeneity and Misalignment Between Experts.** Unlike federated learning, the proposed approach (as well as other mixture/ensemble approaches) generally benefits from heterogeneity, as it results in learning better *local experts* in DCFM Stage 1. Please refer to **Table R3** (resp. to reviewer ToUV).
>
> **Q8.** **Stability Across Decentralized Clients & Necessity for Tuning.** Training is generally stable across clients. However, it is necessary to select an appropriate $\lambda$ in Eqn. 10, 13, 16 (strength of the conditional independence penalty).
>
> **Table R6.** DCFM-A performance, for Colored-MNIST (Non-IID) for various $\lambda$.
>
> | $\lambda$ | FID$^o$ | FID$^*$ |
> | --- | --- | --- |
> | 0.5 | 7.12 | 11.14 |
> | 1 | 7.33 | 9.29 |
> | 10 | 11.82 | 13.65 |
>
> We find that a $\lambda = 1$ or $\lambda = 0.5$ generally work well, while a larger $\lambda=10$ degrades performance. $\lambda = 1$ is used in our experiments.

---

### Official Review · Reviewer_UAfn · 2026-03-12

**Soundness:** 3
**Presentation:** 3
**Significance:** 3
**Originality:** 3
**Overall Recommendation:** 4
**Confidence:** 3

**Summary:**

This work address the challenge of learning a universal generative model from siloed data, and enabling the model to generalise to unseen combinations of discrete conditions. The authors propose a flow matching method, DCFM, that does not require any sample sharing, only sharing of model weights between clients. The method learns a velocity field on each data silo, and enforces each one to respect conditional independence amongst the conditioning variables, before further enforcing that any combination of these fields also respects this conditional independence. They propose two instantiations, DCFM-A, which outputs this set of velocity field estimates to be sampled from in a mixture of experts style, and DCFM-B, which distils the separately learned velocities into a single student model. Both DCFM methods demonstrate improved performance in generation of novel condition combinations compared to existing federated learning and mixture of expert approaches on image generation and robotics planning tasks.

**Compliance With Llm Reviewing Policy:**

Affirmed.

**Final Justification:**

Initially I recommended rejection, because of the heavily lacking experimental details that would be required for the paper to be a useful contribution and be built upon. These have been provided in the rebuttal period, seem reasonable, and improve my impression of the work. Hence I increase my score.

**Key Questions For Authors:**

- Can detailed experimental and training descriptions be provided?
- How were the baseline methods implemented? And were they appropriately tuned?
- What is $w$ in Eq. 11, in the definition of $\mathbf{z}_\theta$?
- Can a flow matching model trained on all the pooled data be evaluated, to assess an upper bound of performance on this data?
- How do the utility scores in Section 6.3 compare to a classifier trained on the real data, again for an upper bound on performance?

**Limitations:**

Yes

**Strengths And Weaknesses:**

**Strengths**
- The paper addresses an important problem, to enable learning generative models in settings where data sharing is infeasible, e.g. for privacy reasons.
- The simple demonstration in Section 4 nicely shows the specific failure mode of previous methods that the paper attempts to address, in generalising to unseen attribute combinations.
- The method is relatively straightforward and explained well.
- The two methods, DCFM-A and DCFM-B, offer different positions along the compute-performance spectrum, with DCFM-A offering better performance, at greater training/inference cost.
- Recent federated learning and mixture of expert flow matching methods are compared against empirically.

**Weaknesses**
- Experimental details are heavily lacking, and, since no code is released either, this significantly hinders reproducibility of the study. No hyperparameters or training details for any method are described in the main text or the appendix, making it impossible to validate results or implement the proposed method.
- No computational comparisons are given with baselines, which may have more simple training schemes than the proposed DCFM method. This makes it difficult to assess the cost/benefit ratio.
- DCFM can only be immediately applied to data with discrete conditioning spaces.
- DCFM is generally outperformed in the metrics for generation with observed combinations of conditions.

**Minor**
- Lines 101-105 are repeated.

---

> ### Author Rebuttal · Authors · 2026-03-31
>
> We thank the reviewer for their useful feedback. We are encouraged that the reviewer finds our problem *important* **(S1)**, our demonstration of the failure modes *clear and empirically supported* **(S2)**, and our method *intuitive and well-explained* **(S3)**. We also appreciate the reviewer’s observation that our proposed variants, DCFM-A and DCFM-B, occupy distinct points on the compute-performance frontier **(S4)**. Finally, the reviewer highlights our empirical comparisons against relevant and recent baselines **(S5)**.
>
> ## Response to Weaknesses & Questions
>
> **(W1, Q1) Training Details.** We will release the code, and provide some additional details:
>
> Common settings:
>
> |  | Mixture-of-2D-Gaussian (Fig. 2) | Colored MNIST | Cube-Single-Play | NIH CXR 14 |
> | --- | --- | --- | --- | --- |
> | # Total Comb. | 4 | 100 | 16 | 54$^*$ |
> | # Observed Comb. | 3 | 50 | 12 | 54$^*$ |
> | # nodes | 2 | 10 | 2 | 4 |
> | Optimizer | AdamW | AdamW | AdamW | AdamW |
> | lr | 1e-4 | 1e-4 | 1e-4 | 1e-4 |
> | # total data | 20K | 300K | 16K | 78K |
> | batch_size | 128 | 128 | 64 | 32 |
>
> Specific Settings:
>
> **(a) Mixture-of-Gaussian (Fig. 2)**
>
> - **Model:**  A small MLP from the **torchcfm** library, specifically `torchcfm.models.MLP`. We also modify the initial layer from `nn.Linear(dim + 1, w)` to `nn.Linear(dim + 1 + k, w)`  to handle multi-attribute conditions.
>
> |  | FL | FL+L | DDM | DDM+L | DFD | DFD+L | DCFM-Stage 1 | DCFM-A | DCFM-B |
> | --- | --- | --- | --- | --- | --- | --- | --- | --- | --- |
> | # epochs / round | 3 | 3 | 30 | 30 | 30 | 30 | 20 | 10 | 30 |
> | # rounds | 10 | 10 | 1 | 1 | 1 | 1 | 1 | 1 | 1 |
> | $\lambda$ (Eqn 10,13,16) | 0 | 1 | 0 | 1 | 0 | 1 | 1 | 1 | 1 |
>
> **(b) Colored MNIST.**
>
> - **Model:**  We use the `UNet2DModel` architecture from the **diffusers** library, with `block_out_channels=[64, 128]` . Some minor modifications are made to the class embedding to accept k=2 conditions instead of 1.
>
> |  | FL | FL+L | DDM | DDM+L | DFD | DFD+L | DCFM-Stage 1 | DCFM-A | DCFM-B |
> | --- | --- | --- | --- | --- | --- | --- | --- | --- | --- |
> | # epochs / round | 1 | 1 | 50 | 50 | 50 | 50 | 40 | 10 | 50 |
> | # rounds | 50 | 50 | 1 | 1 | 1 | 1 | 1 | 2 | 1 |
> | $\lambda$ (Eqn 10,13,16) | 0 | 1 | 0 | 1 | 0 | 1 | 1 | 1 | 1 |
>
> **(b) Cube-Single-Play**
>
> - **Model:**  We use the `TemporalUNet` architecture used by Janner et al. 2022 (Diffuser) and Ajay et al. 2023 (Decision Diffuser).
>
> |  | FL+L | DDM+L | DFD+L | DCFM-Stage 1 | DCFM-A | DCFM-B |
> | --- | --- | --- | --- | --- | --- | --- |
> | # epochs / round | 1 | 100 | 100 | 80 | 10 | 100 |
> | # rounds | 100 | 1 | 1 | 1 | 2 | 1 |
> | $\lambda$ | 1 | 1 | 1 | 1 | 1 | 1 |
>
> **(d) NIH-Chest-XRay-14**
>
> - **Model:**  Latent space: `diffusers.AutoencoderKL` (from Stable Diffusion 2.1), Flow model: `diffusers.UNet2DModel` , with `block_out_channels=[64, 128, 256, 256]`.
>
> |  | FL+L | DDM+L | DFD+L | DCFM-Stage 1 | DCFM-A | DCFM-B |
> | --- | --- | --- | --- | --- | --- | --- |
> | # epochs / round | 1 | 200 | 200 | 120 | 20 | 200 |
> | # rounds | 200 | 1 | 1 | 1 | 4 | 1 |
> | $\lambda$ | 1 | 1 | 1 | 1 | 1 | 1 |
>
> **(W2) Computational Comparison.** Even if allocated more training iterations, baselines fail to improve on novel combinations. Conversely, DCFM effectively leverages additional compute to enhance compositional learning.
>
> **(W3) Only Discrete Conditioning.** We agree that continuous attributes are an important extension. However, as defined in Sec. 3.1, this work focuses solely on discrete attributes, with the aim of clearly elucidating the problem and providing a principled solution. We believe the methods and insights presented in our paper stand as useful contributions on their own.
>
> **(W4)** **Performance for observed combinations.** We agree; however, the slight performance drop on observed combinations is significantly outweighed by the gains in novel compositions. For example, in Tab. 1, while the DDM baseline achieves a slightly better FID-o (8.19 vs. 8.53 for DCFM-A), DCFM-A provides a much stronger FID-novel (11.41 vs. 25.19).
>
> **(Q2) Tuning baselines:** The baselines share the same underlying model architecture and require minimal tuning.
>
> **(Q3) Definition of w:** $w$ represents the classifier-free guidance weight (i.e. from Eqn. 5, if all $w_i$ are equal). In our experiments, we set $w=1$, i.e. no CFG is used in the conditional independence losses.
>
> **(Q4, Q5) Upper Bounds:**
>
> **Table R1.** Centralized models for ColoredMNIST, with (+L) and without a CI penalty.
>
> | Method | FID$^o$ | FID$^*$ |
> | --- | --- | --- |
> | Centralized | **4.71** | 14.3 |
> | Centralized+L | 5.82 | **8.64** |
>
> *NIH-CXR-14:* If we train a classifier on the *real training data*, we find that it achieves a utility score of **0.48**, which is significantly higher than what we obtain (**0.23** in Fig. 5, right). We attribute this performance gap to training exclusively on synthetic data (and testing on real).
>
> ---
>
> Please consider updating your score if we have adequately addressed your concerns.

---

> > ### Author Rebuttal · Reviewer_UAfn · 2026-04-01
> >
> > I thank the authors for their helpful response. Adding these architectural and training details to the paper is a very necessary modification to ensure reproducibility and allow others to build on this work, so I am pleased they have been provided, and hope they will be included in the revised paper. This would largely address my main concern.
> >
> > However, I do have a follow up about computational costs. The authors write:
> > >"Even if allocated more training iterations, baselines fail to improve on novel combinations. Conversely, DCFM effectively leverages additional compute to enhance compositional learning."
> >
> > While this may be the case, it would still be useful to quantify the additional computation required to train DCFM, as opposed to the baselines, to contextualise the cost/benefit of the method. As the authors already state in the manuscript, the adoption of federated diffusion models is hindered by:
> > > "high computational and communication overheads".
> >
> > While some discussion of communication is given in the appendix, the computational aspect is neglected. The authors do write:
> > > "in terms of GPU-hour utilization during training, we empirically observe that DCFM-A is at least 2× as expensive as DCFM-B."
> >
> > Is it possible to offer similar comparisons to the baselines used?

---

> > > ### Author Response · Authors · 2026-04-02
> > >
> > > We thank the reviewer for their response, and are glad that their primary concern (W1) has been largely resolved. We also assume that most of the other concerns and questions (W3,4 & Q1-5) have been addressed.
> > >
> > > **Reviewer’s follow up on computational costs (W2):** We agree with the reviewer that a more detailed quantification and discussion of the computational costs would be useful.
> > >
> > > Given the variability in GPU hours or clock times, we approach computational costs in terms of effective forward/backward passes required during training. We group the methods having similar cost, and provide an average of their FID-novel metric (FID$^*$) from Table 1 (averaged over IID vs Non-IID, and over methods).
> > >
> > > **Table RR:** Cost per local training step of various methods vs. average FID$^*$ (from Table 1 in the paper), where B represents the batch size.
> > >
> > > | Method | # parallel nodes | Forward | Backward | Avg. FID$^*$ |
> > > | --- | --- | --- | --- | --- |
> > > | (a) Classic flow matching (FL, DDM, DFD) | n | 1 x B | 1 x B | 25.33 |
> > > | (b) With local CI penalty (FL+L, DDM+L, DFD+L, DCFM-Stage-1) | n | ~4 x B | ~4 x B | 20.24 |
> > > | (c) Cross-peer penalty (DCFM-A) | n | ~4 x B | ~2 x B | 10.35 |
> > > | (d) Teachers + local CI penalty (DCFM-B) | 1 | ~5 x B | ~4 x B | 10.69 |
> > >
> > > First, for arbitrary number of attributes $k$, practical implementations of conditional independence penalties would require **a constant 4x cost**, by taking an expectation over pairs of two attributes, i.e., $\mathbb{E}\_{(i,j \neq i)\in 1 \dots k}\left[ \left\\| v\_\theta(x\_t \mid y\_i, y\_j) - \left(v\_\theta(x\_t \mid y\_i) + v\_\theta(x\_t \mid y\_j) - v\_\theta(x\_t)\right) \right\\|\_2^2 \right]$.
> > >
> > > From Table RR we observe:
> > >
> > > - **(a)** The *local cost* of federated learning (FL), and the mixture baselines DDM and DFD are exactly the same, as they follow standard flow matching training.
> > > - **(b)** If these baselines are allowed to use local CI penalties (Eqn. 10), their FID$^*$ improves from **~25 to ~20** on average, at the cost of roughly **4x compute / local step**.
> > > - **(c, d)** DCFM-A uses a *StopGrad* operation (Eqn. 12), which reduces the cost in the backward pass. DCFM-B has a similar per-step cost to (a), but requires 1 extra forward pass to obtain teacher predictions. On average, the DCFM methods improve the FID$^*$ from **~25 to ~10,** also approximately at **4x compute / local step**.
> > >
> > > **Empirical Observations:** *While keeping the batch size constant*, compared to the baselines in (a) in Tab. RR, their counterparts in (b) roughly require 2-3x **wall clock time** (different from GPU hours). Compared to DDM and DFD in (a), DCFM-A (c) requires **~4x** wall clock time, while DCFM-B (d) requires roughly **~7x** wall clock time. In practice, to reduce the cost in terms of time, the batch size may be reduced by a factor of 4. In our experiments, we keep the batch size constant for a fair comparison of performance.
> > >
> > > **On DCFM-A using more *GPU hours*:** Though DCFM-A has a cheaper backward pass and may require less effective training time than DCFM-B, it involves learning $n$ different models. This results in higher overall GPU hours (or actual *energy consumption*) than DCFM-B, which trains just 1 model. DCFM-B practically uses less GPU hours, but requires more wall clock time due to the lack of parallelization.
> > >
> > > ---
> > >
> > > We will add the above discussion, as well as a detailed theoretical and empirical analysis of both computational and communication cost of our method and baselines to the appendix of the revised version of the manuscript.

---

### Official Review · Reviewer_swmX · 2026-03-13

**Soundness:** 3
**Presentation:** 3
**Significance:** 3
**Originality:** 3
**Overall Recommendation:** 4
**Confidence:** 4

**Summary:**

This paper proposes a Decentralized Compositional Flow Matching (DCFM) framework for learning generative models with compositional generalization capabilities from centralized private data. It argues that standard decentralized approaches fail in compositional generalization primarily because they do not enforce global conditional independence among generative factors. DCFM addresses this issue through two variants: DCFM-A directly imposes cross-node conditional independence (CI) constraints between local expert models; DCFM-B distills collective knowledge from local experts into a single student model via synthetic data replay and CI penalty mechanisms.

**Compliance With Llm Reviewing Policy:**

Affirmed.

**Key Questions For Authors:**

see the weakness section.

**Limitations:**

yes

**Strengths And Weaknesses:**

**Strength**

- The problem holds practical value, as decentralized learning under privacy constraints grows increasingly important.

- The paper's logic and solution approach are fluid and compelling.

- The supplementary data and experiments across multiple domains are persuasive.



**Weakness**

- The paper’s significance is limited by the scale of the experiments. The largest-scale experiment (chest X-rays) uses only 256×256 images and a client count of n=4. Can DCFM’s cross-node CI penalty mechanism be elegantly scaled to n>>10?

- The mask distribution p(m) suffers from specification issues. Section 5.1 defines the mixed weights $π_{full}$, $π_{marg}$, and $π_{uncond}$ but fails to specify their actual values. These hyperparameters influence the frequency of marginal labels during training, directly determining the effectiveness of the confidence interval penalty.

---

> ### Author Rebuttal · Authors · 2026-03-31
>
> We thank the reviewer for their positive feedback. We are delighted that the reviewer finds our problem practical and of increasing importance (S1), our method logical and compelling (S2), and our experiments and supplemental derivations convincing (S3).
>
> ## Response to Weaknesses & Questions
>
> **(W1) Scaling to n >> 10.** We scale the colored MNIST experiment to $n=20$ Non-IID clients.
>
> **Table R2.** Our method vs. baselines for $n=10$ vs. $n=20$ clients on Colored MNIST.
>
> | Method | n | FID$^o$ | FID$^*$ |
> | --- | --- | --- | --- |
> | DDM | 10 | 7.46 | 17.98 |
> | DCFM-A | 10 | 7.33 | 9.29 |
> | DCFM-B | 10 | 8.49 | 9.15 |
> | DDM | 20 | 7.13 | 24.15 |
> | DCFM-A | 20 | 8.01 | 11.63 |
> | DCFM-B | 20 | 7.52 | 8.98 |
>
> We observe that both the Decentralized Diffusion (DDM) baseline and DCFM-A lose some performance in FID-novel (FID$^*$) as the number of clients double, while DCFM-B shows a slight improvement. In general, the DCFM methods can be scaled to larger number of clients, without any significant loss in performance.
>
> **(W2) Mixture weights.** First, the concept of weights $\pi\_{full}, \pi_{marg}, \pi\_{uncond}$ are largely inherited from classifier-free guidance, which requires training with weights $(\pi\_{full},  \pi\_{uncond})$, commonly set to (0.8, 0.2) or (0.9, 0.1). We find $(\pi\_{full}, \pi\_{marg}, \pi\_{uncond}) = (0.8, 0.1, 0.1)$ to work well in our experiments.

---

> > ### Author Rebuttal · Reviewer_swmX · 2026-04-03
> >
> > Thank you for your responses. The additional n=20 experiment and the mixture weight specification address my concerns. I maintain my score.

---

> > > ### Author Response · Authors · 2026-04-04
> > >
> > > We thank the reviewer for their review and feedback on our work. We will incorporate the additional discussion and ablation in the revised manuscript.

---

### Decision · Program_Chairs · 2026-04-30

**Decision:**

Accept (regular)

**Comment:**

This paper proposes a framework DCFM for training generative models with compositional generalization capabilities from decentralized private data. The authors argue that existing decentralized approaches fail in compositional generalization primarily because they do not enforce global conditional independence among generative factors and compatibility of learned vector fields across local experts. To this end, the proposed approach enforces conditional independence across attributes and aligns locally trained flow models to enable compositional generalization to unseen attribute combinations. The authors propose two variants: DCFM-A, a MoE-style method and a distilled global student model. Both DCFM variants demonstrate improved performance in generation of novel condition combinations compared to existing federated learning and mixture of expert approaches on image generation and robotics planning tasks.

All four reviewers have reached to a positive consensus on the motivation and the novel problem: “The problem holds practical value, as decentralized learning under privacy constraints grows increasingly important” (Reviewer swmX); “The paper addresses an important problem, to enable learning generative models in settings where data sharing is infeasible, e.g. for privacy reasons” (Reviewer UAfn); “The paper addresses an important problem of learning generative models from decentralized datasets with compositional structure” (Reviewer jQNd); “Well-motivated problem” (Reviewer ToUV).

The theoretical foundation and technical approach are well built upon a series of existing works on compositional generative models. Reviewers also found the technical approach is solid: “The paper's logic and solution approach are fluid and compelling” (Reviewer swmX); “The paper proposes a technically sound framework for decentralized compositional generative modeling based on flow matching with conditional independence constraints” (Reviewer jQNd).

Major concerns raised by the reviewers are mainly around experiments and empirical studies, including scale of datasets, simple benchmarks, influence of hyperparameters, training details, ablation study, etc. Most of these concerns have been addressed in the rebuttal with complementary experimental results. Reviewers swmX, UAfn and ToUV have acknowledged that their concerns have been addressed (Reviewer jQNd did not response to the rebuttal).

This paper considers an interesting problem setting, that is FL as a promising scenario for compositional generative models based on conditional independence assumption. The approach itself is solid; however, it inherits some intrinsic unsolved problems from previous compositional generative models (e.g. “only discrete conditioning” and “conditional independence assumptions that may not always hold in realistic settings”), which have also been pointed out by Reviewers UAfn and jQNd. The authors are suggested to either find a suitable real-world FL scenario that fits the current model capability to validate its effectiveness or further improve the model in dealing with continuous and dependent factors in the future.